# Five year mortality in an RCT of a lung cancer biomarker to select people for low dose CT screening

Francis Michael Sullivan[1]*, Frances S. Mair[2], William Anderson[3], Cindy Chew[4], Alistair Dorward[5], John Haughney[6], Fiona Hogarth[7], Denise Kendrick[8], Roberta Littleford[9], Alex McConnachie[2], Colin McCowan[1], Nicola McMeekin[2], Manish Patel[10], Petra Rauchhaus[7], Fergus Daly[1], Lewis Ritchie[11], John Robertson[8], Joseph Sarvesvaran[5], Herbert Sewell[12], Thomas Taylor[13], Shaun Treweek[14], Kavita Vedhara[15], Stuart Schembri[3]

1 University of St Andrews, North Haugh, St Andrews, United Kingdom, 2 Institute of Health & Wellbeing, University of Glasgow, Glasgow, United Kingdom, 3 Respiratory Medicine, NHS Tayside, Dundee, United Kingdom, 4 Radiology, NHS Lanarkshire, Bothwell, United Kingdom, 5 Respiratory Medicine, NHS Greater Glasgow and Clyde, Glasgow, United Kingdom, 6 General Practice, NHS Greater Glasgow and Clyde, Glasgow, United Kingdom, 7 Tayside Clinical Trials Unit, University of Dundee, Dundee, United Kingdom, 8 School of Medicine, University of Nottingham, Nottingham, United Kingdom, 9 Centre for Clinical Research, University of Queensland, Brisbane, Australia, 10 Respiratory Medicine, NHS Lanarkshire, Bothwell, United Kingdom, 11 The Institute of Applied Health Sciences, University of Aberdeen, Aberdeen, United Kingdom, 12 School of Life Sciences, University of Nottingham, United Kingdom, 13 Radiology, NHS Tayside, Dundee, United Kingdom, 14 Health Services Research Unit, University of Aberdeen, Aberdeen, United Kingdom, 15 School of Psychology, Cardiff University, Cardiff, United Kingdom

* fms20@st-andrews.ac.uk

**Data Availability Statement:** Data can be accessed by applying to the ECLS data access committee on the template provided at https://eclsstudy.org/request-for-samples-and-data/.

## Abstract

The role of biomarkers in risk-based early detection of lung cancer may enable screening to become cost effective and widely accessible. EarlyCDT-Lung is an example of such a blood-based autoantibody biomarker which may improve accessibility to Low dose Computed Tomography (LDCT) screening for those at highest risk. We randomized 12 208 individuals aged 50–75 at high risk of developing lung cancer to either the test or to standard clinical care. Outcomes were ascertained from Register of Deaths and Cancer Registry. Cox proportional hazards models were used to estimate the hazard ratio of the rate of deaths from all causes and lung cancer. Additional analyses were performed for cases of lung cancer diagnosed within two years of the initial test. After 5 years 326 lung cancers were detected (2.7% of those enrolled). The total number of deaths reported from all causes in the intervention group was 344 compared to 388 in the control group. There were 73 lung cancer deaths in the intervention arm and 90 in the controls (Adjusted HR 0.789 (0.636, 0.978). An analysis of cases of lung cancer detected within 2 years of randomization in the intervention group showed that there were 34 deaths from all causes and 29 from lung cancer. In the control group there were 56 deaths with 49 from lung cancer. In those diagnosed with lung cancer within 2 years of randomization the hazard ratio for all cause mortality was 0.615 (0.401,0.942) and for lung cancer 0.598 (0.378, 0.946). Further large-scale studies of the role of biomarkers to target lung cancer screening, in addition to LDCT, are likely to provide additional value.

**Funding:** The Scottish Government Health and Social Care Directorate and Oncimmune. Funding information for this article has been deposited with the Crossref Funder Registry.

**Competing interests:** F.M. Sullivan reports grants from Oncimmune and the Scottish Government Health and Social Care Directorate of the Chief Scientist Office, during the conduct of the study. F.S. Mair reports grants from Oncimmune and the Scottish Government Health and Social Care Directorate of the Chief Scientist Office, during the conduct of the study. W. Anderson has nothing to disclose. C. Chew has nothing to disclose. A. Dorward has nothing to disclose. J. Haughney has nothing to disclose. F. Hogarth reports grants from the Scottish Government Health and Social Care Directorate of the Chief Scientist Office and from Oncimmune, during the conduct of the study. D. Kendrick has nothing to disclose. R. Littleford reports grants from the Scottish Government Health and Social Care Directorate of the Chief Scientist Office and Oncimmune, during the conduct of the study. A. McConnachie reports grants from Oncimmune and the Scottish Government Health and Social Care Directorate of the Chief Scientist Office, during the conduct of the study. C. McCowan has nothing to disclose. N. McMeekin reports grants from Oncimmune and the Scottish Government Health and Social Care Directorate of the Chief Scientist Office, during the conduct of the study. M. Patel has nothing to disclose. P. Rauchhaus reports grants from Oncimmune and the Scottish Government Health and Social Care Directorate of the Chief Scientist Office, during the conduct of the study. L. Ritchie has nothing to disclose. J. Robertson reports other funding from Oncimmune, during the conduct of the study; and other funding from Oncimmune, outside the study. J. Robertson was a founder of Oncimmune, a company spun out from the University of Nottingham based on his academic research. Between 2003 and 2013 he was Chief Scientific Officer of Oncimmune and a Director of the company. During this time, he was responsible for the original drafting of the ECLS protocol. Since 2013 he has had no involvement in the science or management of the company. He has been and remains a shareholder in the company. J. Sarvesvaran has nothing to disclose. H. Sewell reports other funding from Oncimmune, outside the submitted work; and was an external member of the Oncimmune Scientific Advisory Board from 2006 to 2013. T. Taylor reports grants, nonfinancial support and other funding from Oncimmune, grants and personal fees from the Chief Scientist Office for Scotland, and grants and nonfinancial support from the Scottish Government, outside the

## Introduction

Lung cancer is commonly diagnosed at a late stage, when five-year mortality rates remain over 90% [1,2]. Methods that detect lung cancer at an earlier stage in those at higher risk, when treatment is more likely to be effective are a priority for many countries [3]. Following the landmark National Lung Screening Trial (NLST) LDCT screening has been shown to detect cancers earlier and reduce lung cancer mortality by 20–25% [4–6]. The US Preventive Task Force currently recommends annual screening with LDCT for adults aged 50 to 80 years who have a 20 pack-year smoking history and currently smoke or have quit within the past 15 years [7]. However, uptake by the public can be suboptimal because of lack of health insurance or low income, difficulties getting time off work, and low perceived risk [8,9]. Moreover, the wider uptake of LDCT screening is limited in many health systems by access to equipment and trained staff, high false positive rates and concerns about overdiagnosis [10–12]. Consequently, in 2023 the American Lung Association reported less than 5% of eligible individuals in the United States had undergone screening [13]. If a more specific, acceptable, less expensive and less resource intensive investigation, such as a biomarker, could precede or accompany imaging, then more people at risk of lung cancer might benefit through better targeted screening or case finding [14,15].

More recent approaches to the evaluation of protein biomarkers in cancer screening emphasise the need to move away from observational studies to trials in complex systems [16]. Few blood-based biomarker tests have proceeded beyond the early stages of biomarker evaluation [17,18]. The EarlyCDT-Lung Test is an enzyme-linked immunosorbent assay (ELISA) that measures seven autoantibodies, each with individual specificity for the following Tumor Associated Antigens [19]. In early, prospective studies it showed a specificity of 91% and sensitivity of 37–41% with better performance at earlier stages of the disease [20]. It has also been evaluated in the management of pulmonary nodules reporting that a positive EarlyCDT-Lung in the presence of a lung nodule is associated with a three-fold increase in the likelihood that the nodule is a lung cancer [21,22]. The ECLS trial was a phase 4 (screening trial) trial powered to determine whether EarlyCDT reduced late stage presentation after two years [23]. A significant reduction in late stage presentation was demonstrated, with a hazard ratio for stage III/IV presentation of 0.64 (95% CI 0.41–0.99), but no significant difference in lung cancer or all-cause mortality at 2 years follow-up. This paper presents a five year follow up per protocol analysis on lung cancer and all cause mortality.

## Methods

### Study design and participants

ECLS was a screening trial which recruited 12 208 high-risk participants through family practices and community-based recruitment strategies in Scotland as described in more detail elsewhere [24,25]. Box. Inclusion and exclusion criteria

Recruitment occurred between August 2013 and June 2016 with follow up undertaken for 60 months after randomization for each participant. This report follows the CONSORT 2010 recommendations and Aarhus guidelines for the reporting of studies on early cancer diagnosis [26,27].

Adults aged 50–75 considered at increased risk of developing lung cancer compared to the general population using established recommendations were eligible to participate [28]. Potential trial participants were identified using electronic medical records of family physicians whose practices were located in the most socioeconomically deprived quintile in Scotland. Some self-referred based on word of mouth or the media advertising used by the study [29]. Ethical board approval was provided by the East of Scotland Research Ethics Committee (REC 13/ES/0024). Written consent was provided by study subjects.

submitted work. S. Treweek reports grants from Oncimmune and the Scottish Government Health and Social Care Directorate of the Chief Scientist Office, during the conduct of the study. K. Vedhara has nothing to disclose. S. Schembri reports grants from Oncimmune and the Scottish Government Health and Social Care Directorate of the Chief Scientist Office, during the conduct of the study.

## Randomisation and masking

After providing informed consent, participants provided a blood sample prior to computer-based randomisation Intervention arm participants a were tested with the EarlyCDT-Lung test. Participants and clinicians were unmasked once the result of the test was known. Mortality. pathology and tumour staging reports were created by independent assessors as part of routine clinical care. The pathologists and register staff were not aware of the allocation status of participants.

## Procedures

LDCT images from test-positive participants were assessed by a panel of experienced thoracic radiologists and respiratory physicians at baseline and every six months afterwards for 24 months or until an abnormality suspected of being lung cancer was detected. Participants in were then followed-up within the study or via the Fleischner society guidelines in the NHS care pathway in operation at that time based on the multidisciplinary team's assessment of which was more appropriate [30]. Participants allocated to the control group, and those who were test negative, had no further study investigations and received standard clinical care available in the UK at that time following National guidelines for identification and management of symptoms suggestive of lung cancer [31]. During the recruitment phase and two years of patient follow up compliance to the protocol was monitored by monthly study team meetings reviewing data derived from the Trial Management Software, OpenClinica, and reports provided by the multidisciplinary teams providing care to study subjects undergoing investigation.

Blood samples were processed according to the trial protocol and Standard Operating Procedures, consistent with relevant UK and US guidelines using a validated method [32]. Quality controlled data on cancer occurrence, mortality and morbidity was obtained from National Services Scotland. These were linked to baseline data collected in Qualtrics in the Dundee Health Informatics Centre Safe Haven [33].

Staging data were taken from the Scottish Cancer Registry (SMR06). Outcome variables extracted from SMR06 were the first occurrence of all diagnoses starting with the International Statistical Classification of Diseases and Related Health Problems 10th Revision codes (ICD-10). If two or more tumours were present at diagnosis, the highest stage determined classification. Pathology reports for other cancers were reviewed to exclude the possibility that such tumours signified lung metastases from a distant primary. Lung tumour histology was coded in accordance with the Third Edition International Classification of Diseases for Oncology and lung cancer staging was determined using TNM 7th Edition [34]. We checked national prescribing, and in- and out-patient data systems for activity relating to trial participants in the five-year follow-up period.

## Statistical analysis

The primary analyses compared all-cause and lung cancer specific mortality between the intervention and control groups. The analyses followed the intention to treat principle. One participant who withdrew consent for use of their data in public records (but not baseline data) was excluded from analysis. The models were adjusted for age, gender, smoking history, and practice. Where models converged, random cluster effects for family practice were included to account for cluster effects. Cox proportional hazard models were used to estimate the hazard ratio. An analysis of proportional hazards by year was performed to test the validity of the model.

Similar methodology was used to analyse the secondary outcomes of mortality rates. Further analysis compared the outcomes of those in the intervention group and those in the control group diagnosed with lung cancer within 2 years of the test—i.e. lung cancers diagnosed within the timescale of the primary endpoint (i.e. reduction in late stage disease at 2 years from randomisation). Comparisons of proportions were carried out using Fisher's exact test due to the small number of events. Poisson regression models, (adjusting for follow up time when necessary) were used to investigate other clinical outcomes.

### Role of the funding source

The funder of the study had no role in study design, data collection, data analysis, data interpretation, or writing of the report.

## Results

### Characteristics of the participants

77 077 invitation letters were sent to people from 166 family practices in our sampling frame and 16 268 responded (21.1%). In addition, 2 389 potential participants in the three participating health boards presented themselves in response to advertising or word of mouth. A telephone interview to confirm eligibility resulted on 12 241 invitations to an in-person screening appointment. 12 215 participants were randomised, 6 were then excluded for the reasons in the CONSORT diagram (Fig 1) and one person withdrew their consent, leaving 12,208 for the analysis. The recruitment rate of people identified as potential study participants from family practice records was 13.4% (10 352/77 077); and the recruitment rate from self-referral was 79.1% (1 889/2 389). Participant characteristics were similar in the intervention and control groups (Table 1). 28.5% (3 477/12 208) of participants lived in the most deprived and a further 23.3% in the 2nd most deprived quintile. The mean age at recruitment was 60.5 years (S.D. 6.58), with a mean pack years smoked of 38.2 (S.D. 18.58). We achieved 99.9% end-point ascertainment in the intervention and control groups.

A similar number of cases of lung cancer were detected in the Intervention group (157) and controls (169) though fewer cases of late stage disease were detected in the intervention group. (S1 Appendix Lung cancer stages at diagnosis by year) The results of all-cause and lung cancer mortality analyses are presented in Table 2A and Fig 2A and 2B. There were 344(5.7%) all cause deaths in the test group compared to 388(6.3%) from all causes in the group not tested 5 years post-randomization. There were 73 lung cancer deaths (1.2%) in the group tested and 90 (1.5%) in the control group. These differences were not statistically significant.

The results for those diagnosed within two years of randomisation are shown in Table 2B and Fig 3A and 3B. There were 34 all-cause deaths (0.6%) and 29 lung cancer deaths (0.5%) in the group who were tested compared to 56 all-cause deaths (0.9%) and 49 lung cancer deaths (0.8%) in the group not tested. When adjusted for age, gender, smoking history, and practice the hazard ratio for all-cause mortality is 0.610 (0.398–0.934, p = 0.0228) and for lung cancer the hazard ratio is 0.598 (0.378, 0.946, p = 0.0281) i.e. in the 10% of people who tested positive and were then diagnosed within two years (Figs 4 and 5). There was a significant reduction in the hazard of lung cancer death.

## Discussion

We have presented five-year follow-up data from an RCT comparing a single autoantibody test (EarlyCDT-Lung) compared to standard clinical practice over five years in a pragmatic study design. The main findings, based on a per-protocol analysis, are that, after five years, all

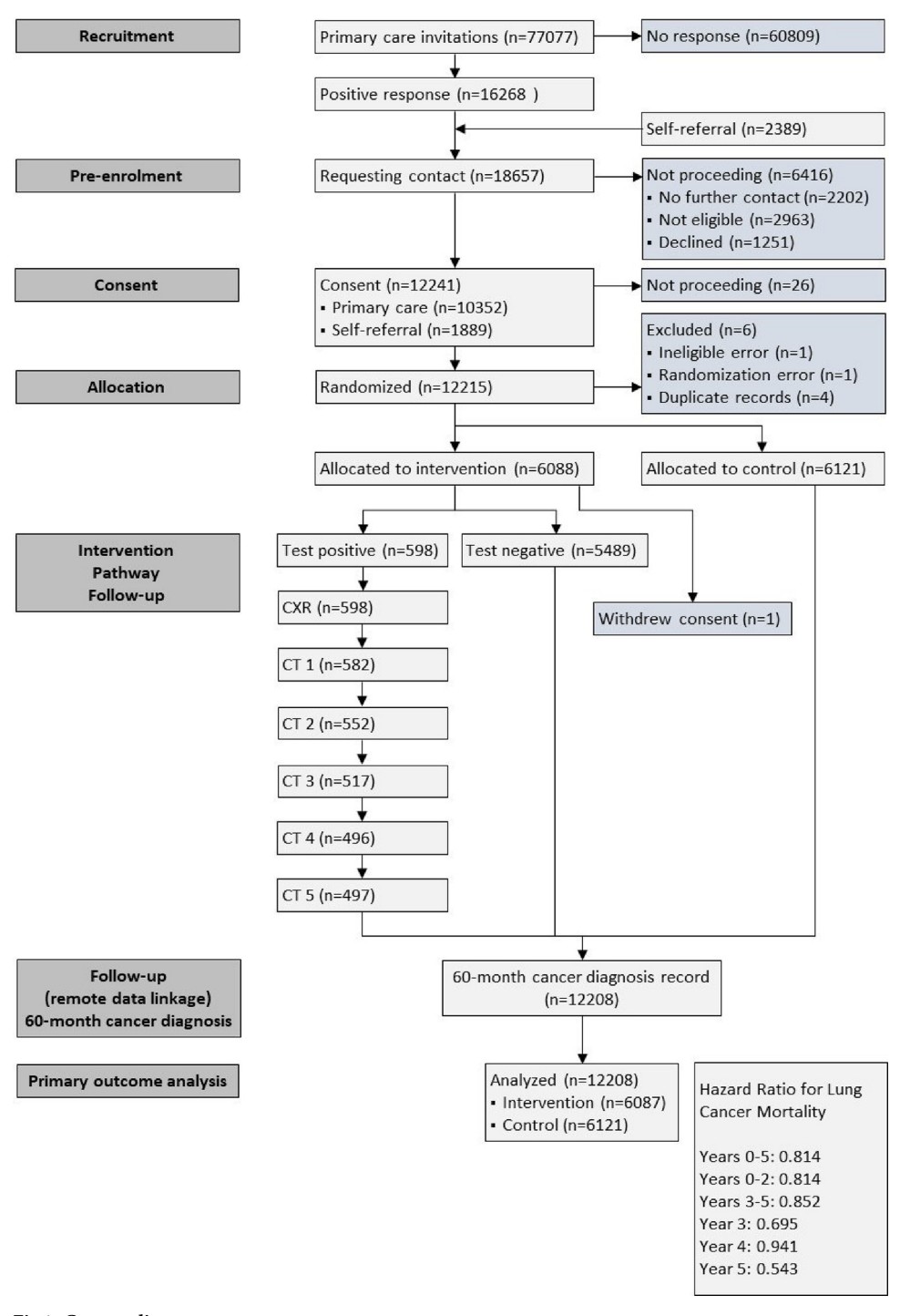

**Fig 1. Consort diagram.**

cause and lung-cancer specific mortality are significantly reduced in patients tested for autoantibodies and diagnosed with lung cancer within two years of the test. The autoantibodies detected by EarlyCDT-Lung are potentially most valuable for detecting early-stage disease in the first year or two after testing and the cancers detected in this study, in those who tested positive, were mainly early stage when patients were able to benefit from recent advances in the management of early stage lung cancer [35].

**Table 1. Characteristics of participants at study entry.**

| Variable | No Test (N = 6121) | Test (N = 6088) |
|---|---|---|
| | N (%) | N (%) |
| **SMID Quintiles** | 60 (1.0) | 50 (0.8) |
| No Information | 2634 (43.0) | 2656 (43.6) |
| Quintile 1 | 1147 (18.7) | 1199 (19.7) |
| Quintile 2 | 770 (12.6) | 736 (12.1) |
| Quintile 3 | 763 (12.5) | 734 (12.1) |
| Quintile 4 | 747 (12.2) | 713 (11.7) |
| Quintile 5 | | |
| **Sex** | | |
| Male | 3129 (51.1) | 3095 (50.8) |
| Female | 2992 (48.9) | 2993 (49.2) |
| **Smoking Status** | | |
| Current Smoker | 3178 (51.9) | 3199 (52.5) |
| Ex-Smoker | 2943 (48.1) | 2889 (47.5) |
| **Age [category]** | | |
| 50–54 years | 1409 (23.0) | 1393 (22.9) |
| 55–59 years | 1531 (25.0) | 1562 (25.7) |
| 60–64 years | 1318 (21.5) | 1300 (21.4) |
| 65–69 years | 12.03 (19.7) | 1179 (19.4) |
| 70–75 years | 660 (10.8) | 654 (10.7) |
| **Age (years)** | 60.5 (6.60) | 60.0 (55.0, 66.0) |
| **Pack year history** | 38.0 (18.47) | 35.0 (25.0, 45.0) |

**Table 2. a. Descriptive Summary Outcomes 5 years post randomization in all patients. b. All Cause & Lung Cancer mortality in patients diagnosed within 2 years of Early CDT.**

| | No Test | | Test | | Total | |
|---|---|---|---|---|---|---|
| Variable | N | (%) | N | (%) | N | (%) |
| Any cause death reported in the 5 years since randomisation? | | | | | | |
| No | 5733 | (93.7%) | 5743 | (94.3%) | 11476 | (94.0%) |
| Yes | 388 | (6.3%) | 344 | (5.7%) | 732 | (6.0%) |
| Total | 6121 | (100.0%) | 6087 | (100.0%) | 12208 | (100.0%) |
| Lung cancer death reported in the 5 years since randomisation? | | | | | | |
| No | 6031 | (98.5%) | 6014 | (98.8%) | 12045 | (98.7%) |
| Yes | 90 | (1.5%) | 73 | (1.2%) | 163 | (1.3%) |
| Total | 6121 | (100.0%) | 6087 | (100.0%) | 12208 | (100.0%) |
| | No Test | | Test | | Total | |
| Variable | N | (%) | N | (%) | N | (%) |
| Any cause death reported in the 5 years since randomisation? | | | | | | |
| No | 6065 | (99.1%) | 6053 | (99.4%) | 12118 | (99.3%) |
| Yes | 56 | (0.9%) | 34 | (0.6%) | 90 | (0.7%) |
| Total | 6121 | (100.0%) | 6087 | (100.0%) | 12208 | (100.0%) |
| **Any Lung Cancer Mortality Reported?** | | | | | | |
| No | 6072 | (99.2%) | 6058 | (99.5%) | 12130 | (99.3%) |
| Yes | 49 | (0.8%) | 29 | (0.5%) | 78 | (0.6%) |
| Total | 6121 | (100%) | 6087 | (100%) | 12208 | (100%) |

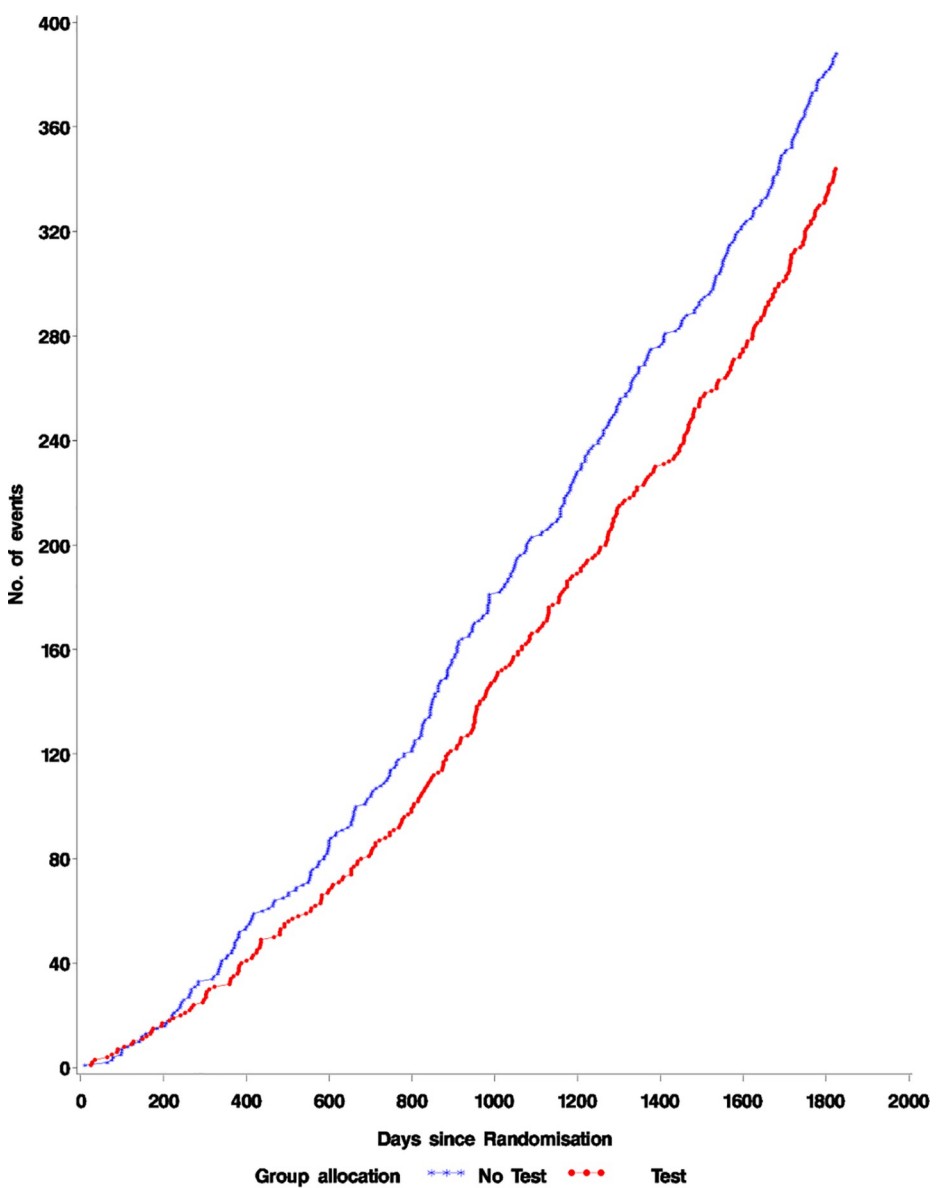

**Fig 2. All cause mortality.**

The cost effectiveness of the test compared to both usual care and LDCT screening for the target population has been published elsewhere [36]. The costs of these two screening methods need to be considered in the context of capacity issues in healthcare services, particularly in UK National Health Service; rolling out LDCT is more resource intensive than a blood test. Finally, during the study, lung cancer participants in the test arm were typically diagnosed at an earlier stage of lung cancer which may be less costly to treat than lung cancer diagnosed at a later stage.

Trials based in the community which are open to a more representative sample of the population are likely to provide results which are more generalisable than those conducted in hospital based studies [37]. In particular, we recruited a high proportion of participants from the two most socioeconomically deprived quintiles (51.8%) a population shown to engage less

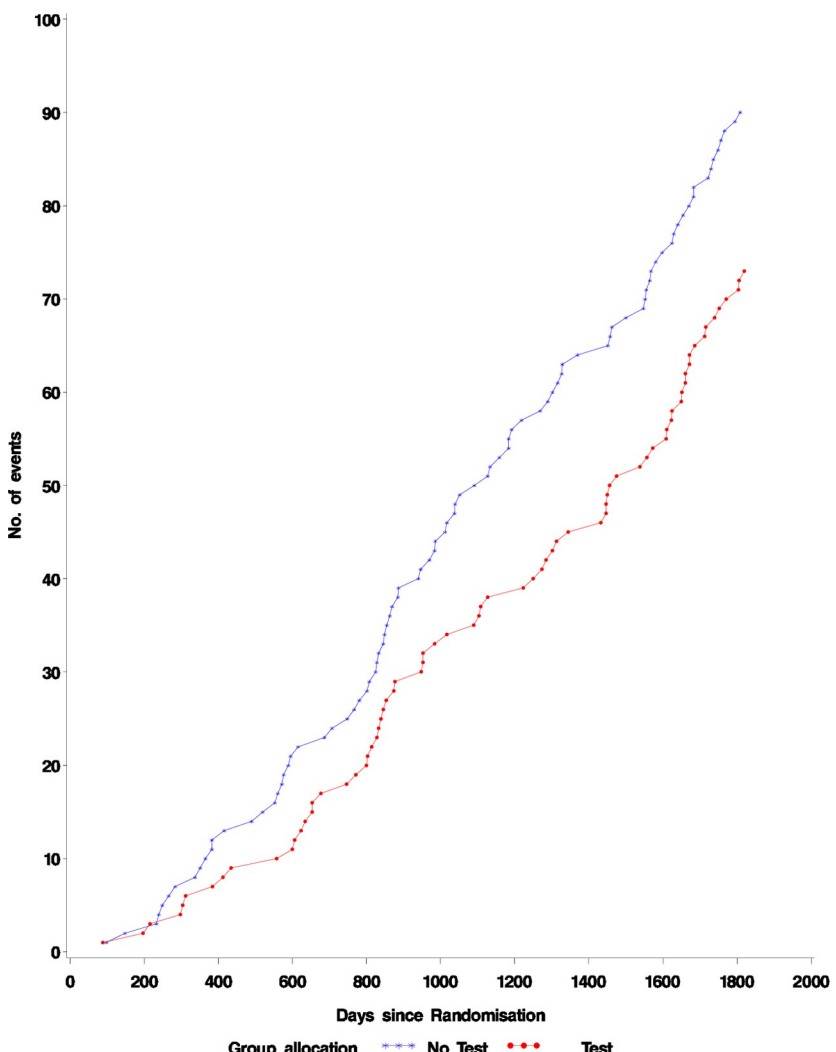

**Fig 3. Lung cancer mortality.**

with lung cancer screening, integration with a national health care system providing whole population care National data linkage enabled a high end-point ascertainment rate (99.9%), and the intention to treat analysis. In addition, the test being investigated (an ELISA) can be performed at relatively low cost in many laboratories, including in countries where LDCT scans are scarce or distributed unequally, increasing the potential for future implementation.

## Limitations

The lung cancer diagnosis rate (2.7%) was lower than we anticipated when planning the study. One potential explanation of this may be the "healthy volunteer" effect, which may have led to a higher rate of recruitment of the healthiest among the high-risk population meeting our inclusion criteria [38]. This is a challenge shared by all research that requires participants to volunteer and provide consent. It is worth noting that even with a lower rate of lung cancer, those in the intervention group were at a statistically significant and clinically important reduced risk of late-stage presentation.

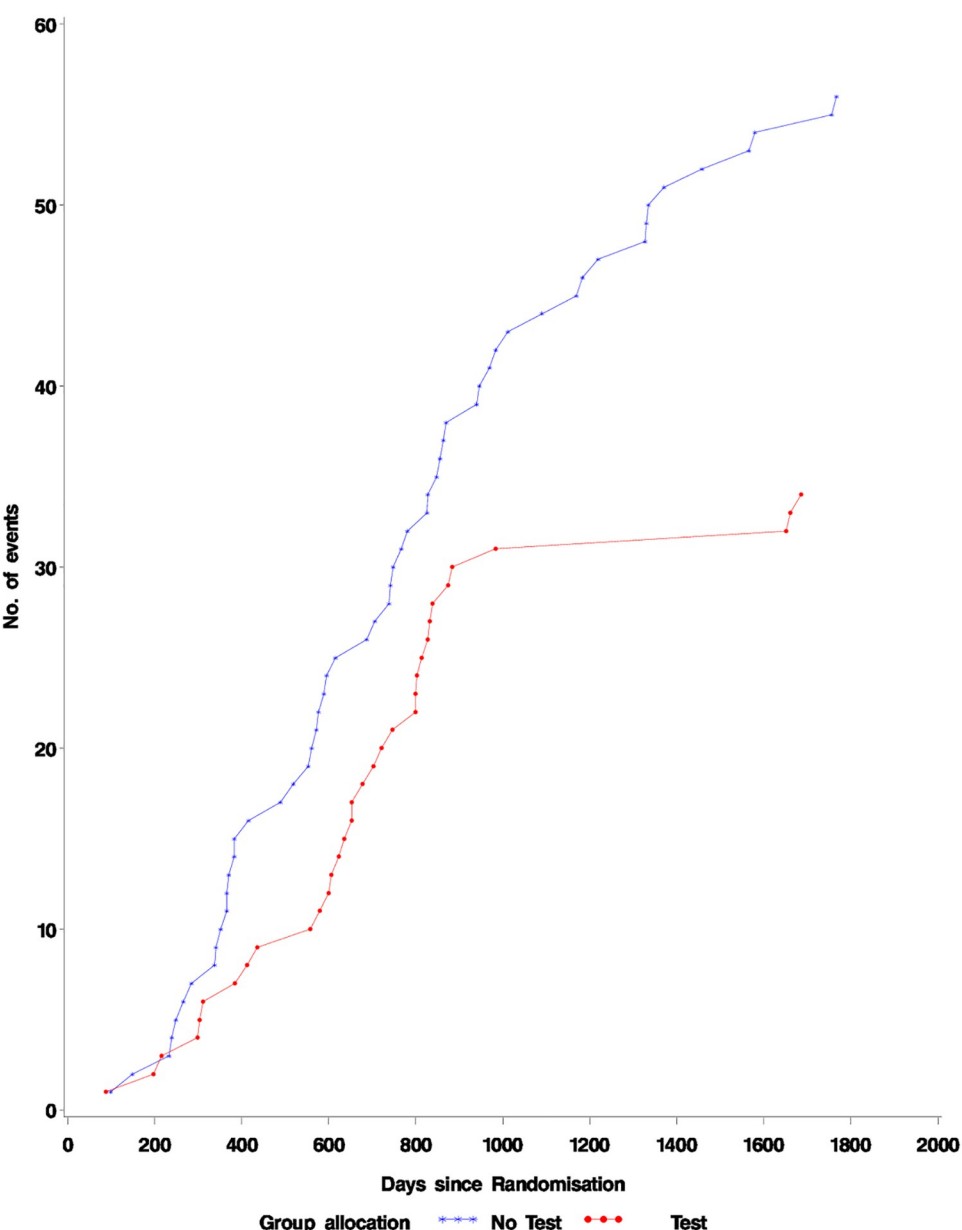

**Fig 4. All cause mortality in patients diagnosed with lung cancer within 2 years of blood test.** Fisher's exact test for a difference in proportions p = 0.025.

The comparator arm used in this RCT, namely awaiting the development of symptoms, is no longer standard clinical practice. Although this does not yet constitute a 'standard of care' many professional organizations recommend that individuals at high risk for lung cancer (i.e. the targeted population in this RCT) should be screened using LDCT if they are potential candidates for curative-intent therapy rather than being observed for the development of symptoms. This situation lowers the clinical impact of the results. A control arm involving CT screening would have provided evidence comparing USPSTF guidelines against a '*biomarker first*' approach, but this was not available when the ECLS trial started and it is still unavailable in many health systems [39]. We plan a ten year follow up and a cost-effectiveness analysis

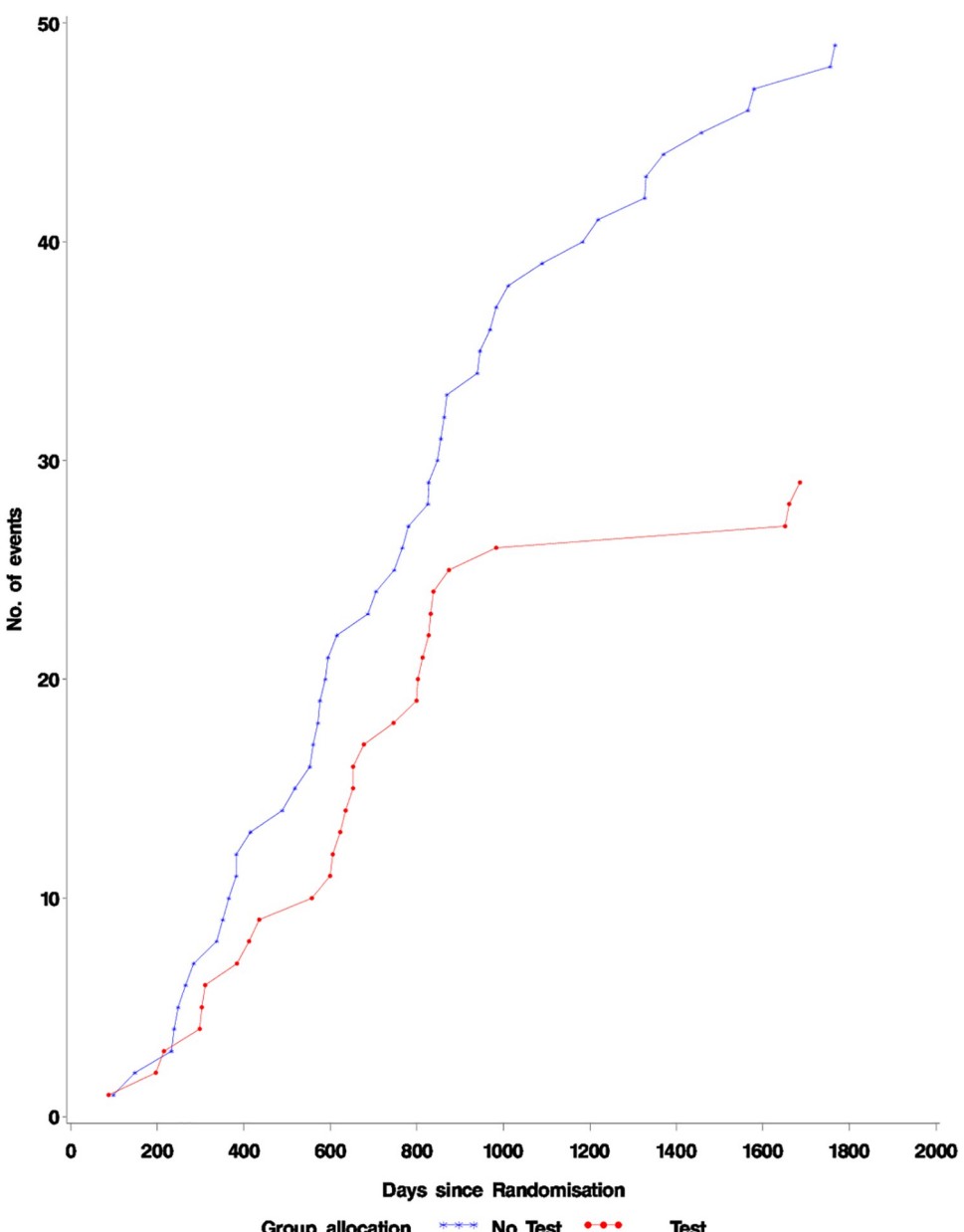

**Fig 5. Lung Ca.** Mortality in patients diagnosed with lung cancer within 2 years of blood test. Fisher's exact test for a difference in proportions p = 0.030.

employing a model to link the short-term outcomes measured within the trial to potential longer-term impacts on health related to morbidity and mortality of early detection and treatment, to allow the estimation of cost-per Quality Adjusted Life Years gained.

Blood tests or other biomarkers could substantially reduce the number of people requiring imaging investigations depending upon where the cut-off for sensitivity and specificity is set. This may have globally significant implications for case finding and screening for lung cancer in people at high risk of the disease. Whether blood based biomarkers should be used in screening, case-finding, or to identify individuals at increased biological risk who might be considered in future prevention approaches, requires further elucidation. The relatively low

sensitivity of Early CDT in this study means that many biomarker negative patients will not have undergone LDCT, this precludes its use as a sole screening method.

The high specificity of the EarlyCDT-Lung test could be used in combination (concurrently or sequentially) with LDCT, which demonstrates high sensitivity, to ensure a high detection rate of early lung cancer cases: a previous report showed that the EarlyCDT-Lung test enhanced the positive predictive power of CT scan and nodule-based risk models for the detection of lung cancer. A highly specific non-invasive investigation to confirm or clarify pathology would also be of significant clinical benefit for often elderly patients with small volume disease for whom obtaining tissue is challenging.

## Conclusions

ECLS demonstrates that a blood-based panel of autoantibodies, as in the EarlyCDT-Lung test, may have an important role in future lung cancer screening programmes. The trial provides proof of concept and clinical utility that blood testing in combination with optimal selection of high-risk people and imaging can find cancers at the earliest stages when they are most amenable to cure. Further investigation in large, community-based phase V studies are needed to determine the long-term impact of performing the EarlyCDT-Lung test on mortality; cost-effectiveness, the level of risk that should be targeted; the optimal time interval between tests, and how to improve the engagement of patients at the highest risk [40].

## Supporting information

**S1 Appendix.**
(TIF)

**S1 File.**
(PDF)

## Author Contributions

**Conceptualization:** Francis Michael Sullivan, Frances S. Mair, William Anderson, Cindy Chew, Alistair Dorward, John Haughney, Fiona Hogarth, Denise Kendrick, Roberta Littleford, Alex McConnachie, Colin McCowan, Nicola McMeekin, Manish Patel, Petra Rauchhaus, Fergus Daly, Lewis Ritchie, John Robertson, Joseph Sarvesvaran, Herbert Sewell, Thomas Taylor, Shaun Treweek, Kavita Vedhara, Stuart Schembri.

**Funding acquisition:** Francis Michael Sullivan.

**Methodology:** Francis Michael Sullivan, Frances S. Mair.

**Project administration:** Francis Michael Sullivan, Frances S. Mair.

**Resources:** Francis Michael Sullivan, Frances S. Mair.

**Supervision:** Francis Michael Sullivan, Frances S. Mair.

**Writing – original draft:** Francis Michael Sullivan.

**Writing – review & editing:** Frances S. Mair, William Anderson, Cindy Chew, Alistair Dorward, John Haughney, Fiona Hogarth, Denise Kendrick, Roberta Littleford, Alex McConnachie, Colin McCowan, Nicola McMeekin, Manish Patel, Petra Rauchhaus, Fergus Daly, Lewis Ritchie, John Robertson, Joseph Sarvesvaran, Herbert Sewell, Thomas Taylor, Shaun Treweek, Kavita Vedhara, Stuart Schembri.

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
