## [Decision Letter · Decision Letter 0]

29 Jul 2024

PONE-D-24-21605Improved five year mortality in an RCT of a lung cancer biomarker to select people for screening.PLOS ONE

Dear Dr. Sullivan,

Thank you for submitting your manuscript to PLOS ONE. After careful consideration, we feel that it has merit but does not fully meet PLOS ONE’s publication criteria as it currently stands. Therefore, we invite you to submit a revised version of the manuscript that addresses the points raised during the review process.

We look forward to receiving your revised manuscript.

Kind regards,

Eugenio Paci, MD

Academic Editor

PLOS ONE

Journal Requirements:

3. Thank you for stating the following in the Competing Interests section: "F.M. Sullivan reports grants from Oncimmune and the Scottish Government Health and Social Care

Directorate of the Chief Scientist Office, during the conduct of the study. F.S. Mair reports grants from Oncimmune and

the Scottish Government Health and Social Care Directorate of the Chief Scientist Office, during the conduct of the

study. W. Anderson has nothing to disclose. 

C. Chew has nothing to disclose. A. Dorward has nothing to disclose. J. Haughney has

nothing to disclose. F. Hogarth reports grants from the Scottish Government Health and Social Care Directorate of the

Chief Scientist Office and from Oncimmune, during the conduct of the study. D. Kendrick has nothing to disclose.

R. Littleford reports grants from the Scottish Government Health and Social Care Directorate of the Chief Scientist

Office and Oncimmune, during the conduct of the study. A. McConnachie reports grants from Oncimmune and the

Scottish Government Health and Social Care Directorate of the Chief Scientist Office, during the conduct of the study.

C. McCowan has nothing to disclose. N. McMeekin reports grants from Oncimmune and the Scottish Government

Health and Social Care Directorate of the Chief Scientist Office, during the conduct of the study. M. Patel has nothing

to disclose. P. Rauchhaus reports grants from Oncimmune and the Scottish Government Health and Social Care

Directorate of the Chief Scientist Office, during the conduct of the study. L. Ritchie has nothing to disclose.

J. Robertson reports other

funding from Oncimmune, during the conduct of the study; and other funding from Oncimmune, outside the

study. J. Robertson was a founder of Oncimmune, a company spun out from the University of Nottingham based on his

academic research. Between 2003 and 2013 he was Chief Scientific Officer of Oncimmune and a Director of the

company. During this time, he was responsible for the original drafting of the ECLS protocol. Since 2013 he has had no

involvement in the science or management of the company. He has been and remains a shareholder in the company.

J. Sarvesvaran has nothing to disclose. H. Sewell reports other

funding from Oncimmune, outside the submitted work; and was an external member of the Oncimmune Scientific

Advisory Board from 2006 to 2013. T. Taylor reports grants, nonfinancial support

and other funding from Oncimmune, grants and personal fees from the Chief Scientist Office for Scotland, and grants

and nonfinancial support from the Scottish Government, outside the submitted work. 

S. Treweek reports grants from Oncimmune and the Scottish

Government Health and Social Care Directorate of the Chief Scientist Office, during the conduct of the study.

K. Vedhara has nothing to disclose. S. Schembri reports grants from Oncimmune and the Scottish Government Health

and Social Care Directorate of the Chief Scientist Office, during the conduct of the study."

Additional Editor Comments:

In this paper, the authors update all-cause and cause-specific mortality at 5 years post-randomization in the ECLES randomized screening trial. The 2-year mortality results, study design, methods, and data on lung cancer diagnoses within 24 months of randomization have already been published. (1) Briefly, the intervention and control groups were randomized to receive LCS biomarker testing or usual care. Subjects positive for the biomarker were selected to receive an intensive screening protocol (with good acceptability): 4 LDCT tests within 2 years of randomization (plus chest x-ray). The update of the stages of lung cancers detected is presented in the appendix of the new text for each year since randomization, for a total of 5 years, confirming the diagnostic anticipation in the biomarker positives. As noted by the statistical editor, the presentation of the all-cause and cause-specific mortality plots lacks statistical information, which is briefly reported in the text.

I think, as noted reviewer 1, there is confusion in the analysis and, in my opinion, ambiguity in the title and the presentation of the results. The 5-year improvement in lung cancer mortality cannot be attributed to lung cancer biomarker selection: positive subjects had an intensive protocol of LDCT testing within 2 years which was shown beneficial. There are two combined interventions, biomarker testing for subjects’ selection and LDCT for screening of the selected subjects. In a paper referenced in the text, this has been defined as multimodal screening because the aim is to improve the screening performance and cost. (2)

In the previous ECLES paper, the sensitivity of the biomarker was estimated to be 31.2% at 2 years. The proportion of positive subjects in the intervention group was approximately 10%, and at 2 years, 18 lung cancer cases were diagnosed in the positive intervention group (N=598) and 38 in the negative intervention group (N=5489). At 5 years, the total number of positive and negative cases was 28 and 129, respectively. In terms of population impact, the low sensitivity of the biomarker means that there are lung cancer cases that are not selected by the biomarker but would have benefited from screening in terms of better staging if selected. There is a benefit in the biomarker group compared to negatives, but the benefit is diluted because of the low sensitivity of the biomarker. The absolute number of Lung Cancers not selected by the biomarker must be better understood (time between diagnosis and biomarker test?).

Thus, the study confirmed the efficacy of LDCT screening in biomarker-positive cases, but the selection process is conditioned on the limited performance of the EarlyCDT-Lung test. The selection using a biomarker is finalized to control costs (fewer people screened), but a population-based screening is expected to have high population coverage, i.e. offer diagnostic anticipation to a high fraction of lung cancers in the population. I agree with the authors that further studies are needed. However, I suggest reconsidering the complexity of this multimodal screening process when revising the paper for the new version.

1) Sullivan FM, et al. Early Diagnosis of Lung Cancer Scotland (ECLS) Team. Earlier diagnosis of lung cancer in a randomised trial of an autoantibody blood test followed by imaging. Eur Respir J. 2020 Jul 30:2000670. doi:

10.1183/13993003.00670-2020.

2) Carozzi, F. M. et al. Multimodal lung cancer screening using the ITALUNG biomarker panel and low dose computed tomography. Results of the ITALUNG biomarker study. Int. J. Cancer 141, 94–101 (2017).

Reviewers' comments:

Reviewer's Responses to Questions

**Comments to the Author**

1. Is the manuscript technically sound, and do the data support the conclusions?

Reviewer #1: Partly

Reviewer #2: Partly

2. Has the statistical analysis been performed appropriately and rigorously? 

Reviewer #1: Yes

Reviewer #2: No

3. Have the authors made all data underlying the findings in their manuscript fully available?

Reviewer #1: Yes

Reviewer #2: Yes

4. Is the manuscript presented in an intelligible fashion and written in standard English?

Reviewer #1: Yes

Reviewer #2: Yes

5. Review Comments to the Author

Reviewer #1: the paper presents important results from a larg and well-conducted RCT. The authors highlight the limits of the study honestly. I think the importance of the paper is even higher than the authors mostly recognize.

Abstract

First sentence: it is not clear if this sentence is a will or a normative statement. It sounds strange for a research paper.

Methods

Please report the inclusion criteria for high-risk.

In the objective, a per-protocol analysis is announced, nevertheless, in the methods I could not understand how the compliance to the protocol was defined and how the non-compliant participants were treated. I miss a definition of the comparisons proposed in the analyses.

I suggest to compare the outcomes in the model including a variable that classifies the participants in three groups according to randomization arm and test result. According to the intervention design, those in the intervention arm but testing negative should not have any advantage of screening. Thus, classifying the participants in 3 groups, the control arm, the intervention testing negative and the intervention testing positive, the model should show similar adjusted hazards for the control and the intervention arm testing negative, and a reduced hazard only for those testing positive. If the hazard in the testing negative is different from that in the control arm it means that the adjusting variables are not sufficient to correct for the selection effect of testing in the intervention arm or that a different baseline risk was present after randomization (this difference is unlikely to be large, nevertheless it seems that a certain imbalance is present).

I also suggest reporting the results in the flow chart with yearly risk by arm and testing results.

Results

Last sentence, I do not understand what it means “i.e. in the 10% of people who tested positive and were then diagnosed within two years.” Did you compare the Hazard in those who teste positive vs. the entire control arm? If this is the mean of pr-protocol analysis, I think it is a too extensive interpretation…

Discussion

In the main findings, please explain the intervention: who underwent the test (people at high risk, heavy smokers? Judging from the average pack/y I guess, but this info is never given in the paper), the management of those positive to the test (yearly LDCT for 5 years).

Discussion should focus on the reliability and limits of the “per-protocol” analysis which is very questionable. N important point to discuss is the consistency of the findings by test results: it is clear an early diagnosis effect with 15 cancers found in the 10% who tested positive in the first year followed by an average of 3 cancers per year in this group.

Reviewer #2: PONE-D-24-21605: statistical review

Summary. This is a five-year follow-up study where the authors test whether all-cause and lung-cancer specific mortality are reduced in patients tested for autoantibodies and diagnosed with lung cancer within two years of the test. The statistical analysis relies on a battery of Cox and Poisson regressions. While the study seems well conducted, the presentation of the results is quite poor: see my major concerns below. I also append some specific points that should be addressed.

Major points.

1. Too little is said about the results obtained by the Cox regression analysis. I’d welcome a traditional table with estimates and standard errors where we do not only see the estimate of the treatment effect but also the coefficients of the confounding variables (age, gender, smoking history, and practice). This is not only important for a better interpretation of the results, but it also helps results reproducibility.

2. The sentence “Where models converged, random cluster effects for family practice were included to account for cluster effects” is a bit obscure. It looks like the paper includes models with and without a cluster effect. But where are the results of these models?

3. Similarly, on page 6 the authors mention Poisson regression models. However, I did not find any Poisson regression analysis in the paper …

Specific points.

1. Section “Outcomes” should be entitled “Statistical analysis” . this section should also include the significance level that has been used to reject null hypotheses.

2. Page 5: the sentence “Cox proportional hazard models were used to estimate the hazard ratio” is repeated twice.

3. Page 6: “An analysis of proportional hazards by year was performed to test the validity of the models”. The authors should report the results of the proportionality test as supplementary material.

4. Page 7: “These differences were not statistically significant.” The p-value should be reported.

6. PLOS authors have the option to publish the peer review history of their article (what does this mean?). If published, this will include your full peer review and any attached files.

Reviewer #1: **Yes: **Paolo Giorgi Rossi

Reviewer #2: No

---

## [Author Response · Author response to Decision Letter 0]

4 Sep 2024

Editor’s requests

1. Ensure that your manuscript meets PLOS ONE's style requirements, including those for file naming 

a. Done

2. The grant information you provided in the ‘Funding Information’ and ‘Financial Disclosure’ sections do not match 

a. Done

3. Include the following statement: ""This does not alter our adherence to PLOS ONE policies on sharing data and materials.” (as detailed online in our guide for authors http://journals.plos.org/plosone/s/competing-interests

a. Added to submission template and updated cover letter.

4. Include your updated Competing Interests statement in your cover letter

a. Done

5. Include your full ethics statement in the ‘Methods’ section of your manuscript file. In your statement, please include the full name of the IRB or ethics committee who approved or waived your study, as well as whether or not you obtained informed written or verbal consent. 

a. Done

6. Please include captions for your Supporting Information files at the end of your manuscript, and update any in-text citations to match accordingly 

a. Done

7. Update title to reduce ambiguity

a. Change made

Reviewer 1

8. Amend 1st sentence of abstract

a. Change made

9. Report the inclusion criteria for high-risk .

a. Provided in references 24 and 25. If required a box containing the text in supplementary material section 9 could be added at the point on page 4 indicated in the text,

10. Clarify how the compliance to the protocol was defined 

a. Added to relevant section of text with greater detail in reference 24

11. Compare the outcomes in the model including a variable that classifies the participants in three groups according to randomization arm and test result. 

a. See section 11 of the supplementary material which deals with this point. Although this subgroup analysis may be of interest to readers, we have not included it in the main paper as the study is presenting the results of a trial comparing the Intervention V Control.

12. Report the results in the flow chart with yearly risk by arm & testing results. 

a. Hazard ratios have been added to the consort diagram or the table presented in the Supplementary material section 

13.Provide data comparing the Hazard (ratio) in those who tested positive vs. the entire control arm. 

b. See Supplementary table section 13

14 Explain the intervention: who underwent the test (people at high risk, heavy smokers 

c. See response 9 above

15. Discussion should focus on the reliability and limits of the “per-protocol” analysis 

a. Changes made to paragraph 1 of the discussion to emphasise this point

Reviewer 2

16. Provide a table with estimates and standard errors where we do not only see the estimate of the treatment effect but also the coefficients of the confounding variables (age, gender, smoking history, and practice) 

a. Added as section 16 to supplementary material

17. Provide the models for when models converged 

a. Provided in response to 16 above

18. Provide details of Poisson regression analysis 

a. See supplementary material section 18

19. Change title of Outcomes section to Statistical analysis

a. Done

20. Remove repeated sentence on p5

a. Done

21. Report the results of the proportionality test as supplementary material.

a. See Supplementary material section 21 

22. Report p values on p7 

a. done 

Additional editorial requests

23. Upload your figure files to the Preflight Analysis and Conversion Engine (PACE) digital diagnostic tool, https://pacev2.apexcovantage.com/

Done though see point 12

---

## [Decision Letter · Decision Letter 1]

2 Oct 2024

PONE-D-24-21605R1Five year mortality in an RCT of a lung cancer biomarker to select people for Low Dose CT screening.PLOS ONE

Dear Dr. Sullivan,

Thank you for submitting your manuscript to PLOS ONE. After careful consideration, we feel that it has merit but does not fully meet PLOS ONE’s publication criteria as it currently stands. Therefore, we invite you to submit a revised version of the manuscript that addresses the points raised during the review process.

We look forward to receiving your revised manuscript.

Kind regards,

Eugenio Paci, MD

Academic Editor

PLOS ONE

Additional Editor Comments:

Thank you for resubmitting and editing the manuscript. The comments of the reviewers received sufficient responses in your revision, and I feel the paper is now much more informative, especially in Supplemental material. However, I confirm my editorial judgment to the authors. My comment was about the sensitivity of the biomarker used in this study and the number of lung cancer cases in the population not screened because negative for the biomarker test. Your analysis for protocol, reported in the supplemental material offers information that is not presented in the main text and the discussion section.

Looking at the 3 groups (Biomarker positive, negative, and control group), after 5 years you have an LC incidence of 28 (4.7%), 129 (2.4%), and 169 2.8%. There is diagnostic anticipation (3.2%) in biomarker positive, but the number of LC biomarker negative subjects without the benefit of screening (negative for the biomarker test) was large.

Looking at “Any LC death reported” mortality is similar in the three groups (1.3, 1.2, 1,5, respectively). In Table 13 the Hazard ratio for Lung cancer mortality comparing positive vs negative at 5 years showed HR 0.91 (0.0442-1.88) and negative vs controls 0.80 (0.58-1.10). The HR was modified adjusting for several variates but never statistically significantly. This can be expected, in RCTs the follow-up since the start is usually at 10 years to see a LC mortality reduction. The issue is the positive and negative tested subjects at 5 years did not show evident differences between and limited when all are compared with the controls (and possibly more attributable to the negative subjects) .

In the Discussion session, you stated “The main findings, based on a per-protocol analysis, are that, after five years, all-cause and lung cancer-specific mortality were significantly reduced in patients tested for autoantibodies and diagnosed with lung cancer within two years of the test.” You should be explicit about the estimates on which this statement was based.

You should include the results of the protocol analysis in the main text, also the figures can be better understood if they are considered in the graphs. In the legend, the numbers which are supporting the evidence should be reported.

Detail: I cannot understand the number of supplemental tables starting at 9.

Reviewers' comments:

Reviewer's Responses to Questions

**Comments to the Author**

1. If the authors have adequately addressed your comments raised in a previous round of review and you feel that this manuscript is now acceptable for publication, you may indicate that here to bypass the “Comments to the Author” section, enter your conflict of interest statement in the “Confidential to Editor” section, and submit your "Accept" recommendation.

Reviewer #1: All comments have been addressed

Reviewer #2: All comments have been addressed

2. Is the manuscript technically sound, and do the data support the conclusions?

Reviewer #1: Yes

Reviewer #2: (No Response)

3. Has the statistical analysis been performed appropriately and rigorously? 

Reviewer #1: Yes

Reviewer #2: (No Response)

4. Have the authors made all data underlying the findings in their manuscript fully available?

Reviewer #1: Yes

Reviewer #2: (No Response)

5. Is the manuscript presented in an intelligible fashion and written in standard English?

Reviewer #1: Yes

Reviewer #2: (No Response)

6. Review Comments to the Author

Reviewer #1: I find the supplementary material more interesting and informing than the main analyses...

Nevertheless the whole paper is an important step in lung cancer screening

Reviewer #2: (No Response)

7. PLOS authors have the option to publish the peer review history of their article (what does this mean?). If published, this will include your full peer review and any attached files.

Reviewer #1: **Yes: **Paolo Giorgi Rossi

Reviewer #2: No

---

## [Author Response · Author response to Decision Letter 1]

16 Oct 2024

Dear Dr. Paci, 

Thank you, once again, we have made the changes requested and respond to the numbered points made below. We hope these are satisfactory.

1. My comment was about the sensitivity of the biomarker used in this study and the number of lung cancer cases in the population not screened because negative for the biomarker test. 

a. Additional text added to the limitations section of the discussion (p9) to emphasise this fact ‘The relatively low sensitivity of Early CDT in this study means that many biomarker negative patients will not have undergone LDCT, this precludes its use as a sole screening method.’

2. Your analysis for protocol, reported in the supplemental material offers information that is not presented in the main text and the discussion section.

a. We agree, but have provided the additional material at the request of reviewers rather than as the core text of the article. Readers who wish to consider more detailed analyses can access this additional material.

3. …the number of LC biomarker negative subjects without the benefit of screening (negative for the biomarker test) was large.

a.We hope this is now adequately addressed in response to point 1 above

4. …in RCTs the follow-up since the start is usually at 10 years to see a LC mortality reduction.

a. We plan a 10 year follow up, but consider the five year findings sufficiently important to current debates to merit publication sooner than 10 years.

5. The main findings, based on a per-protocol analysis, are that, after five years, all-cause and lung cancer-specific mortality were significantly reduced in patients tested for autoantibodies and diagnosed with lung cancer within two years of the test.” You should be explicit about the estimates on which this statement was based.

a.The information is presented in table 2b and the relevant results section on p7. 

See also amended figures 3 a and 3b which now provide the Fisher’s exact test results.

6. You should include the results of the protocol analysis in the main text, also the figures can be better understood if they are considered in the graphs.

a.The five year per-protocol analysis of all study participants is provided in the second section of the results and refers to Table 2a and as graphs in Figures 2a and 2b.

7. In the legend, the numbers which are supporting the evidence should be reported.

a. See response to 5a, the legends for amended figures 3 a and 3b provide the Fisher’s exact test results.

8. I cannot understand the number of supplemental tables starting at 9.

a. The numbers refer to our responses to previous referee’s requests. The numbering starts at 9 as this was the 9th response.

9. I find the supplementary material more interesting and informing than the main analyses...

Nevertheless the whole paper is an important step in lung cancer screening

a. Thank you, we think that most readers will prefer the main text, but some, like the reviewers, will find the supplementary material more interesting.

Please do not hesitate to contact me should you require further information.

---

## [Decision Letter · Decision Letter 2]

31 Oct 2024

Five year mortality in an RCT of a lung cancer biomarker to select people for Low Dose CT screening.

PONE-D-24-21605R2

Dear Dr. Sullivan,

We’re pleased to inform you that your manuscript has been judged scientifically suitable for publication and will be formally accepted for publication once it meets all outstanding technical requirements.

Kind regards,

Eugenio Paci, MD

Academic Editor

PLOS ONE

Additional Editor Comments (optional):

Reviewers' comments:

Reviewer's Responses to Questions

**Comments to the Author**

1. If the authors have adequately addressed your comments raised in a previous round of review and you feel that this manuscript is now acceptable for publication, you may indicate that here to bypass the “Comments to the Author” section, enter your conflict of interest statement in the “Confidential to Editor” section, and submit your "Accept" recommendation.

Reviewer #2: All comments have been addressed

2. Is the manuscript technically sound, and do the data support the conclusions?

Reviewer #2: (No Response)

3. Has the statistical analysis been performed appropriately and rigorously? 

Reviewer #2: (No Response)

4. Have the authors made all data underlying the findings in their manuscript fully available?

Reviewer #2: (No Response)

5. Is the manuscript presented in an intelligible fashion and written in standard English?

Reviewer #2: (No Response)

6. Review Comments to the Author

Reviewer #2: (No Response)

7. PLOS authors have the option to publish the peer review history of their article (what does this mean?). If published, this will include your full peer review and any attached files.

Reviewer #2: No

---

## [Editor Report · Acceptance letter]

7 Nov 2024

PONE-D-24-21605R2 

PLOS ONE

Dear Dr. Sullivan, 

I'm pleased to inform you that your manuscript has been deemed suitable for publication in PLOS ONE. Congratulations! Your manuscript is now being handed over to our production team.

Kind regards, 

on behalf of

Dr. Eugenio Paci 

Academic Editor

PLOS ONE